# A General Solution to the Continuum Rate Equation for Island-Size Distributions: Epitaxial Growth Kinetics and Scaling Analysis

**DOI:** 10.3390/nano15050396

**Published:** 2025-03-04

**Authors:** Vladimir G. Dubrovskii

**Affiliations:** Faculty of Physics, St. Petersburg State University, Universitetskaya Emb. 13B, 199034 St. Petersburg, Russia; dubrovskii@mail.ioffe.ru

**Keywords:** rate equations, island-size distribution, continuum limit, scaling

## Abstract

The nucleation and growth of surface islands in the pre-coalescence stage has previously been studied by different methods, including the rate equation approach and kinetic Monte Carlo simulations. However, full understanding of island growth kinetics and the scaling properties of their size distributions is still lacking. Here, we investigate rate equations for the irreversible homogeneous growth of islands in the continuum limit, and derive a general island-size distribution whose shape is fully determined by the dynamics of the monomer concentration at a given size dependence of the capture coefficients. We show that the island-size distribution acquires the Family–Viscek scaling shape in the large time limit if the capture coefficients are linear in size for large enough islands. We obtain analytic solutions for the time-dependent monomer concentration, island density, average size and island-size distribution, which are valid for all times, and the analytic scaling function in the large time limit. These results can be used for modeling growth kinetics in a wide range of systems and shed more light on the general properties of the size distributions of different nano-objects.

## 1. Introduction

The nucleation and growth of “clusters”, including surface islands of different types [1,2,3,4,5,6,7], droplets [8] and vertical wires [9,10,11], have been studied using the rate equation (RE) approach [1,3,5,6,7], kinetic Monte Carlo (KMC) simulations [2,4] and other techniques. Modeling island-size distributions (ISDs) in different systems is essential for understanding and fine-tuning material properties at the nanoscale and from the fundamental viewpoint [12]. The ISD is usually considered in terms of the “natural” variables s and θ, where s is the number of monomers in an island (“size”) and θ=It is the coverage, with I as the external flux (deposition rate for surface islands) and t as time. Studies of the ISDs for epitaxial islands in the pre-coalescence stage of their irreversible growth (with the critical size of unity) and the ISDs in some extended models [2,3,4,5,13,14,15,16,17,18,19,20,21,22,23,24,25,26,27,28,29,30,31,32,33,34,35,36,37,38,39,40,41,42,43,44] revealed their scaling properties for large enough times or island sizes. The Family–Viscek (FV) scaling [13] requires the appropriately normalized ISD to be time-invariant in terms of the scaled size $x=s/\langle s\rangle$. The ISD scaling in the FV form is anticipated for epitaxial surface islands in the limit of the infinitely large ratio of the adatom diffusion coefficient D over the deposition rate I (Λ=D/I→∞) for all but very short times [3,5,14,15,16,19,22,25,27,32,37,38]. The assumption of the FV shape of ISDs is so usual that solutions to the continuum RE (at s≫1) are often searched in this form [16,19,22,32]. The FV scaling has far-reaching implications in areas far beyond epitaxial island growth [45], and hence is interesting from the general perspective of statistical physics.

It is well-known that the FV-type ISD requires certain properties of the capture coefficients $\sigma_{s}$ [16,22,32]. Size-independent $\sigma_{s}=const$ [14,16] and the power-law $\sigma_{s}\sim s^{\beta }$ with β<1 [22,36] resulted in non-analytic scaling based on the continuum REs, with singularities arising at large x and at x→0 when β>0. Non-analytic scaling is not commonly observed in KMC simulations [3,4,5,16,32] and does not support the FV scaling hypothesis in general. The dependence of $\sigma_{s}$ on the island size s and coverage θ is, however, more complex than in the self-consistent theory [17,18], because in the scaling limit Λ→∞ the islands compete for the adatom diffusion flux in a non-trivial manner [16,32]. Theoretical analysis of the capture zones based on the Voronoi tessellation and direct KMC simulations revealed a linear increase of $\sigma_{s}$ with s at large enough s [16,22,27,32,40]. The slope of $\sigma_{s}$ was found independent of θ for compact island morphologies [32]. Using this property, analytic scaling functions in the FV form were obtained in Refs. [37,38], but only in the large time limit.

On the other hand, the Fokker–Planck-type continuum RE for the ISD was previously studied in classical nucleation theory with a large critical size i≫1. Kuni et al. [8] showed that the continuum ISD is time-invariant in terms of a certain “invariant size” ρ, for which the regular growth rate of different islands is independent of their size. This time-invariance is widely used in the growth modeling of a wide range of nano-objects, including epitaxial islands [6,7,12]. According to Refs. [12,46,47,48], the time invariance of the continuum ISD requires negligible kinetic fluctuations, described by the second derivative with respect to size in the continuum RE. This is valid when $\sigma_{s}\sim s^{\beta }$ with β>1/2 at large s, including the important case of $\sigma_{s}\sim s$ [12,46]. Interestingly, the absence of fluctuation-induced broadening in terms of invariant variables corresponds to a larger variance in the ISD in terms of natural size s [46,47]. In irreversible growth, the shape of the time-invariant ISD is fully determined by the time dependence of the monomer concentration [36]. The relationship between time-invariant ISDs in terms of natural or invariant variables is obviously non-trivial and requires thorough study. Consequently, here we develop the irreversible growth theory and obtain a general solution to the continuum RE that requires no assumptions (apart from s≫1) and is valid for any time and at any Λ. Next, we analyze the rare cases where the time-invariant solution in terms of invariant size transitions into the FV scaling in terms of natural size s. We also present an exactly solvable model for size-linear${\;\sigma }_{s}$, which can be useful for understanding the general properties of ISDs and the growth modeling of different nano-objects.

## 2. General Considerations

We start from the discrete set of REs describing the irreversible island growth:(1)$$\frac{dn_{1}}{d\theta }=1-2\Lambda n_{1}^{2}-\Lambda n_{1}\sum_{s=2}^{\infty }\sigma_{s}n_{s},\;\frac{dn_{s}}{d\theta }=\Lambda n_{1}(\sigma_{s-1}n_{s-1}-\sigma_{s}n_{s}),\;s\geq 2,$$
with the initial conditions(2)$$n_{s}\left(0\right)=0\;\mathrm{for}\;\mathrm{all}\;s.$$
Here, θ=It, Λ=D/I and $\sigma_{1}=1$. According to Equation (1), the monomer concentration $n_{1}$ increases due to deposition, and decreases due to the attachement of monomers to islands of all sizes with s≥2 and dimerization. The surface concentration of islands $n_{s}\;$for any s≥2 increases due to the attachment of monomers to islands having size s−1 and decreases due to attachment of monomers having size s. The initial conditions correspond to bare substrate at zero time. Choosing $\sigma_{1}=1$ is justified because the form given by Equation (1) is resumed for any $\sigma_{1}^{0}\neq 1$ and $D_{0}\;$by the simple transformation $D={\sigma_{1}^{0}D}_{0}$ and $\sigma_{s}=\sigma_{s}^{0}/\sigma_{1}^{0}$ for s≥2. We do not consider the direct impingement terms and coalescence [32]. For surface islands, this requires Λ≫1 and θ≪1.

Introducing the normalized concentrations $f_{s}$ and time τ by definitions(3)$$n_{s}=\frac{f_{s}}{\sqrt{\Lambda }},\;\theta =\frac{\tau }{\sqrt{\Lambda }}.$$
Equation (1) is rewritten as(4)$$\frac{df_{1}}{d\tau }=1-2f_{1}^{2}-f_{1}\sum_{s=2}^{\infty }\sigma_{s}f_{s},\;\frac{df_{s}}{d\tau }=f_{1}\left(\sigma_{s-1}f_{s-1}-\sigma_{s}f_{s}\right)\;,\;s\geq 2.$$
The normalized surface density of islands F, the total number of monomers in all islands with s≥2 G and the average size $\langle s\rangle$ are given by(5)$$F=\sum_{s=2}^{\infty }f_{s},\;G=\sum_{s=2}^{\infty }sf_{s},\;\langle s\rangle =\frac{G}{F}.$$
The nucleation rate in irreversible growth is given by [12,32,36,37,38](6)$$\frac{dF}{d\tau }=f_{1}^{2}.$$

The mass balance has the form(7)$$\tau =f_{1}+G=f_{1}+F\langle s\rangle ,$$
showing that, in the absence of desorption, the total number of monomers equals τ. Assuming that the capture coefficients $\sigma_{s}$ are independent of Λ [3,4,5,16,27,32], Equations (4)–(6) contain no Λ dependence. In this case, any solutions to the REs for the ISD $n_{s}$, island surface density N and total number of monomers in the islands Q should have the form(8)$$n_{s}=\frac{1}{\sqrt{\Lambda }}f_{s}(\sqrt{\Lambda }\theta ),\;N=\frac{1}{\sqrt{\Lambda }}F(\sqrt{\Lambda }\theta ),\;Q=\frac{1}{\sqrt{\Lambda }}G(\sqrt{\Lambda }\theta ).$$
In the limit of Λ→∞ typically considered for FV scaling behavior, the islands rapidly reach large sizes in the pre-coalescence growth stage at modest coverages θ≪1.

## 3. General Solution for ISD in the Continuum Limit

In the continuum limit s≫1, the discrete REs given by Equation (4) are reduced to one Fokker–Planck-type equation for f(s,τ). If the capture coefficients σ(s) increase with s faster than $s^{1/2}$ [16,22,27,32], one can safely neglect the second derivative with respect to size in the continuum RE [36]. Following the general method [8,36,46,47], we introduce the invariant variables by definition(9)$$z=\int_{0}^{\tau }d\tau^{’}f_{1}\left(\tau^{’}\right)=\int_{1}^{s_{max}}\frac{ds^{’}}{\sigma (s^{’})},$$(10)$$\rho =\int_{1}^{s}\frac{ds^{’}}{\sigma (s^{’})}.$$
Here, ρ is the invariant size for which the growth rate of all islands is independent of their size and equals unity. The time-dependent invariant size z corresponds to islands that have emerged at zero time. Therefore, it gives the maximum invariant size, which is related to the maximum natural size $s_{max}$. Larger islands cannot be present in the ISD. The nucleation rate becomes(11)$$\frac{dF}{dz}=f_{1}.$$
Here, the monomer concentration $f_{1}$ should be presented as a function of z. The new ISD $g\left(\rho ,z\right)$ over the invariant size ρ is introduced by [36](12)$$f\left(s\right)=\frac{1}{\;\sigma \left(s\right)}g(\rho ).$$

The first order continuum RE is given by [36](13)$$\frac{\partial g}{\partial z}=-\frac{\partial g}{\partial \rho },\;g\left(\rho =0,z\right)=f_{1}(z),$$
and has the solution(14)$$g\left(\rho ,z\right)=f_{1}\left(z-\rho \right),\;0\leq \rho \leq z.$$
This ISD is a function of one variable z−ρ. Hence, it maintains a time-invariant shape at any time [12,36]. Using Equation (12), the ISD over natural size is obtained in the form(15)$$f\left(s,s_{max}\right)=\frac{1}{\sigma \left(s\right)}f_{1}(z-\rho ),$$
where(16)$$z-\rho =\int_{s}^{s_{max}}\frac{ds^{’}}{\sigma (s^{’})}.$$
With the known size-dependent capture coefficients $\sigma \left(s\right)$, this general solution requires no assumptions.

The particular shape of the continuum ISD is fully determined by the z-dependent monomer concentration $f_{1}(z)$. This function should be obtained as a solution to the reduced Equation (4) for $f_{1}$ or, equivalently, from the mass balance given by Equation (7). With the ISD given by Equation (14), the island density and the total number of monomers in the islands become(17)$$F=\int_{1}^{s_{max}}dsf\left(s,s_{max}\right)=\int_{0}^{z}dxf_{1}(x),\;G=\int_{1}^{s_{max}}dssf\left(s,s_{max}\right)=\int_{0}^{z}dxf_{1}\left(x\right)s(z-x).$$
Here, the island size $s\left(\rho \right)$ should be obtained by inverting Equation (10). Inverting Equation (9), we find:(18)$$\tau \left(z\right)=\int_{0}^{z}\frac{dx}{f_{1}(x)}.$$
Using this in Equation (7), the mass balance takes the form(19)$$f_{1}\left(z\right)=\int_{0}^{z}\frac{dx}{f_{1}(x)}-\int_{0}^{z}dxf_{1}\left(x\right)s(z-x).$$
With the known $s\left(\rho \right)$, this non-linear integral equation fully determines the continuum ISD in Equation (15). Its detailed analysis for different capture coefficients σ(s) will be presented elsewhere.

## 4. Family–Viscek Scaling

The FV scaling form of the ISD at Λ→∞ requires that [13](20)$$\frac{{\langle s\rangle }^{2}}{\theta }n\left(s,\theta \right)=h\left(x\right),\;x=\frac{s}{\langle s\rangle }$$
for all but very short times (or for all but very low θ). Here, $h\left(x\right)$ is a universal function of the scaled size x that must satisfy the usual constrains for the island density and coverage:(21)$$\int_{0}^{\infty }dxh\left(x\right)=\int_{0}^{\infty }dxxh\left(x\right)=1.$$
We first note that ${\langle s\rangle }^{2}n\left(s\right)/\theta ={\langle s\rangle }^{2}f(s)/\tau$ in our normalizarion. Then, the mass balance yields $\tau =F\langle s\rangle$ at $f_{1}\to 0$. The island density F saturates to a constant when ${\;\sigma }_{s}\sim s^{\beta }$ with β>1/2: $F\to F_{\infty }=const$ at $f_{1}\to 0$ [27,32,36,37,38]. Furthermore, the average size must be proportional to the maximum size: $\langle s\rangle =ks_{max}$ at $f_{1}\to 0$, with a time-independent k<1. Hence, we arrive at the following asymptotic behavior of the ISD:(22)$$\frac{{\langle s\rangle }^{2}}{\theta }n\left(s,\theta \right)=\frac{\langle s\rangle }{F_{\infty }}f\left(s,\langle s\rangle \right)=\frac{1}{F_{\infty }}\frac{\langle s\rangle }{\sigma \left(s\right)}f_{1}\left[\int_{s}^{\langle s\rangle /k}\frac{ds^{’}}{\sigma (s^{’})}\right],$$
where $f_{1}(z)$ is a solution to Equation (19).

From these considerations, the question of whether the ISD acquires the FV scaling form at $f_{1}\to 0$ is equivalent to the following: can the function defined by Equation (22) be an analytic function of the scaled size $x=s/\langle s\rangle$ for a given $\sigma \left(s\right)$? Non-analytic scaling functions, obtained earlier, for example, in Refs. [14,16,22,36], are not very interesting. Indeed, any solution for an ISD can finally be presented as a non-analytic function of $x=s/\langle s\rangle$ [22,36], but the singularities in the scaled ISD may be misleading and non-physical. Non-analytic scaling functions follow directly from Equation (22) for the power law capture coefficients $\sigma \left(s\right)=s^{\beta }$ with β<1. This was earlier demonstrated in Refs. [22,36]. Leaving aside the general study, we note that the simplest form of the capture coefficients that satisfies the FV scaling hypothesis is their linear increase with s at large enough s:(23)$$\sigma \left(s\right)=s\;\mathrm{for}\;s\gg 1.$$
According to Refs. [16,27,32], the linear scaling of $\sigma \left(s\right)\;$with s is due to the fact that larger islands have larger capture zones for adatoms, which are roughly proportional to the island surface area s for compact island morphologies. For the size-linear capture coefficients, Equation (22) yields the FV scaling function of the form(24)$$h\left(x\right)=\frac{A}{x}f_{1}\left[ln(kx)\right].$$
Unfortunately, however, Equation (19) cannot be solved exactly even with the exponential $s\left(\rho \right)=\exp(\rho )$ for all sizes. It can be reduced to the second order differential equation for the monomer concentration of the form(25)$$f_{1}^{2}\frac{d^{2}f_{1}}{dz^{2}}+\frac{df_{1}}{dz}+f_{1}=0,$$
whose solution is unknown. Therefore, in the next section we consider an exactly solvable model with a simplified $f_{1}(z)$ that meets the asymptotic behaviors $f_{1}\left(z\right)\to \sqrt{2z}$ at z→0 and $f_{1}\left(z\right)\to \exp(-z)$ at z→∞ following from Equation (25) in the general case.

## 5. Exactly Solvable Model

Based on the above considerations, we use the size-linear capture rates given by Equation (23), in which case the invariant variables defined by Equations (9) and (10) become(26)$$\rho =lns,\;z=lns_{max},\;z-\rho =ln(s_{max}/s).$$
The integral term in Equation (4) for $f_{1}\left(\tau \right)$ can be approximated by(27)$$f_{1}\sum_{s=1}^{\infty }\sigma_{s}f_{s}≅f_{1}\sum_{s=1}^{\infty }sn_{s}=f_{1}\tau .$$
While the behavior of the capture coefficients $\sigma_{s}$ at small s can differ significantly from the simple linear law [16,32], the rapid island growth at Λ→∞ should lead to the predominant contribution of large s in the sum $\sum_{s=1}^{\infty }\sigma_{s}f_{s},\;$where the linear approximation of $\sigma_{s}$ becomes accurate. Using Equation (27) in Equation (4), we obtain the closed differential equation for $f_{1}\left(\tau \right)$:(28)$$\mathrm{as}\frac{df_{1}}{d\tau }=1-2f_{1}^{2}-f_{1}\tau .$$
The exact solution to this equation with the initial condition $f_{1}\left(\tau =0\right)=0$ is given by(29)$$f_{1}\left(\tau \right)=\frac{erf(\tau /\sqrt{2})}{\sqrt{2/\pi }e^{-\tau^{2}/2}+\tau erf(\tau /\sqrt{2})},$$
where erf⁡(ξ) is the error function. The asymptotic behaviors of the monomer concentration versus time are $f_{1}\left(\tau \right)\to \tau$ at τ→0 and $f_{1}\left(\tau \right)\to 1/\tau$ at τ→∞. Using Equation (9), we obtain the asymptotic behaviors of the monomer concentration in terms of z: $f_{1}\left(z\right)\to \sqrt{2z}$ at z→0 and $f_{1}\left(z\right)\to exp(-z)$ at z→∞, as in the general case. However, Equation (9) with this $f_{1}\left(\tau \right)$ cannot be integrated analytically to give precisely the required dependence $f_{1}(z)$.

In what follows, we consider a simplified function,(30)$$f_{1}\left(\tau \right)=\frac{\tau }{1+\tau^{2}}.$$
According to Figure 1a, this function approximates the solution given by Equation (29) with reasonable accuracy. Figure 1b shows the scaled monomer concentration as a function of coverage and demonstrates the influence of the parameter Λ. It is clear that the monomer concentration, and hence the nucleation rate, of islands become sharper for larger Λ. In the limit of Λ→∞, the monomer concentration reaches its maximum at a very low θ≪1 and then decreases slowly (as 1/θ or exp⁡(−z)) for a much longer time. Only this long tail of the distribution is usually considered in the scaling limit [14,16,37,38]. In our model, this is not essential, because Equation (29) (or its approximation given by Equation (30)) describe the ISD starting from the very beginning of deposition.

Using Equation (9), we obtain(31)$$z=\frac{1}{2}ln(1+\tau^{2}),\;s_{max}=\sqrt{1+\tau^{2}}.\;$$
Using $\tau =\sqrt{\exp\left(2z\right)-1}$ in Equation (30), we obtain(32)$$f_{1}\left(z\right)={\left(e^{2z}-1\right)}^{1/2}e^{-2z},$$
with the correct asymptotic behaviors at z→0 and z→∞. According to Figure 1a, the simplified function ${\;f}_{1}\left(\tau \right)$ underestimates the monomer concentration, and hence the nucleation rate (which is proportional to $f_{1}^{2}(\tau )$). To account for this mismatch, we introduce a constant α into the nucleation rate: $dF/d\tau =\alpha f_{1}^{2}(\tau )$ or $dF/dz=\alpha f_{1}(z)$, which is equivalent to $g\left(z-\rho \right)=\alpha f_{1}\left(z-\rho \right).$ This α will be determined later in a self-consistent manner. The explicit dependence $f_{1}\left(z\right)$ given by Equation (32) yields the analytic continuum ISD over invariant size:(33)$$g\left(z-\rho \right)=\alpha {\left[e^{2(z-\rho )}-1\right]}^{1/2}e^{-2(z-\rho )}.$$

From Equations (15) and (26), the analytic ISD over natural size is given by(34)$$f\left(s,s_{max}\right)=\frac{\alpha }{s_{max}}\sqrt{1-{\left(\frac{s}{s_{max}}\right)}^{2}}.$$
Using Equation (34), the island density and the total number of monomers in the islands are easily obtained from Equation (17):(35)$$F=\alpha \left[arctg\left(\frac{\sqrt{s_{max}^{2}-1}}{s_{max}+1}\right)-\frac{1}{2}\frac{\left(s_{max}-1\right)}{s_{max}^{2}}\right],$$(36)$$G=\frac{\alpha }{3}s_{max}{\left[1-s_{max}^{-2}\right]}^{3/2}.$$
As expected, the island density saturates at large $s_{max}$: $F\to F_{\infty }=\alpha \pi /4$, while G becomes proportional to $s_{max}$: $G\to (\alpha /3)s_{max}$. According to Equation (31), $s_{max}\to \tau$ at large enough τ. Hence, the average size tends to(37)$$\langle s\rangle =\frac{G}{F}\to \frac{4}{3\pi }s_{max}=\frac{4}{3\pi }\tau$$
regardless of α. The mass balance given by Equation (7) at $f_{1}\to 0$ yields $F_{\infty }\langle s\rangle \to \tau$, while in our model $F_{\infty }\langle s\rangle \to (\alpha /3)\tau$. Hence, we must put(38)α=3
in the above expressions to ensure the correct normalization of the ISD.

Figure 2 and Figure 3 show the continuum ISDs over invariant and natural sizes for different $s_{max}$ from 10 to 200. These distributions have very different shapes. The ISD $g\left(z-\rho \right)$ in Figure 2 maintains its time-invariant shape at any time, because the growth rate of differently sized islands is the same (and equals unity in our normalization). This ISD has a maximum corresponding to the most representative invariant size in the distribution. Islands of this size have emerged at the maximum monomer concentration [8,12,36]. This shape of the ISD is fully determined by the time-dependent monomer concentration in Figure 1a, presented as a function of the variable z according to Equation (32). Conversely, the shape of the ISD $f\left(s,s_{max}\right)$ in Figure 3 undergoes significant changes in the course of growth. It becomes flatter for larger s, similar to the geometrical distribution of probabilities with size-linear rate constants [12]. These ISD have a maximum at s→0, so that small islands remain most representative in the distribution at any time. This property is different from the mono-modal ISDs obtained by KMC simulations [32] and analytical solutions to REs with size-independent capture coefficients [16,36] or with a critical size i>1 [19].

Figure 4 shows the monomer concentration $f_{1}(\tau )$, island density F(τ), maximum size $s_{max}\left(\tau \right)$, average size $\langle s\rangle \left(\tau \right)$ and total number of monomers in the islands G(τ), obtained from Equations (30), (31), (35), (36) and (5). These graphs are given in a logarithmic scale to better show the island growth kinetics in the initial stage, that is, before reaching a maximum of $f_{1}$ at τ~1. This short nucleation stage is usually disregarded in scaling theories, most of which consider only the asymptotic stage at $f_{1}\to 0$ [14,15,22,37,38]. However, it is important in the overall mass balance, because a significant fraction of the islands nucleate at very short times. This is due to the fact that the maximum nucleation rate corresponds to the maximum of the monomer concentration. All curves shown in the figure contain no Λ dependence if expressed in terms of the scaled time $\tau =\sqrt{\Lambda }\theta$. The monomer concentration decreases as 1/τ in the large time interpolation, leading to a slow saturation of the island density in the asymptotic stage. These trends are well-known and have been reproduced by different methods [3,5,32,49]. The total number of monomers in the islands G approaches $s_{max}≅\tau$ soon after reaching the maximum of $f_{1}$. The average size $\langle s\rangle$ is much smaller than $s_{max}$ in the nucleation stage, but rapidly tends to $as_{max}$ in the asymptotic stage.

Using Equation (34), it is easy to obtain the FV scaling function in the form(39)$$\frac{{\langle s\rangle }^{2}f(s)}{\tau }\to h\left(x\right)=3a^{2}\sqrt{1-{\left(ax\right)}^{2}},\;a=\frac{4}{3\pi },\;0\leq x\leq \frac{1}{a}.$$
This function satisfies both normalization conditions given by Equation (21), because(40)$$\int_{0}^{1/a}dxh\left(x\right)=\int_{0}^{1/a}dxxh\left(x\right)=1.$$
The collapse of the scaled ISDs to the universal shape is shown in Figure 5. It is seen that the ISDs in the FV scaling variables become indistinguishable from h(x) starting from $s_{max}\sim 50$. The analytic scaling function given by Equation (39) is monotonically decreasing, has a maximum of $16/(3\pi^{2})≅0.54\;$at x=0 and becomes zero at $x_{max}=3\pi /4≅2.35.$ The analytic scaling functions obtained previously for irreversible homogeneous growth with size-independent capture coefficients σ(s)=const are distinctly different. These functions are monotonically increasing, become discontinuous at a certain $x_{c}>1$ and then tend to zero [14,16,22]. For the power-low capture-coefficients $\sigma (s)=s^{\beta }$ with 0<β<1, the scaling functions are non-monotonic, with two discontinuities at x=0 and $x=x_{c}$ and a minimum between them [22,36]. The analytic scaling functions of the Amar–Family type [19] have a maximum and tend to zero at x→0, but they require a critical cluster with size i>1. The analytic scaling function obtained in Ref. [38] is two-parametric and reproduces both mono-modal and monotonically decreasing shapes depending on the parameters. It applies, however, only to the asymptotic growth stage where $f_{1}≅1/\tau =\exp\left(-z\right).$ The Polya-type FV scaling function [10] applies to heterogeneous nucleation, for example, for vertical nanowires emerging from a fixed number of growth seeds on a substrate surface. KMC simulations lead to continuous scaling functions with a maximum, which are non-zero at x=0 [32]. As discussed, for example, in Refs. [27,32], the maximums of the ISDs and their corresponding FV scaling functions most probably originate from a more complex behavior of the capture coefficients σ(s) at small s rather than the simple approximation $\sigma \left(s\right)=s$ at s≫1. For example, if $\sigma \left(s\right)≅const$ for small $s\ll \langle s\rangle$ and converges to the linear dependence $\sigma \left(s\right)=s$ at $s\sim \langle s\rangle$, one may anticipate a maximum in the ISDs, which is developed in the early stage of growth. These refinements of the model will be considered in a forthcoming paper.

## 6. Conclusions

In summary, we have obtained the general solution to the continuum RE of homogeneous irreversible growth, whose shape is fully determined by the time-dependent concentration of monomers. In contrast to the common approach [16,22], we have considered the continuum RE in terms of invariant size ρ rather than the scaled size $x=s/\langle s\rangle$, and have not used any assumptions on the capture coefficients (for example, their scaling as $\sigma (s)/\langle \sigma \rangle =C(x))$. To provide an analytic solution for the ISD, the dependence $f_{1}\left(\tau \right)$ should be presented as a function of the invariant variable z, which is not a simple problem in the general case. We plan to study in detail the integral Equation (19) using different forms of the capture coefficients $\sigma \left(s\right)$ in a forthcoming work. Special attention should be paid to the coverage-dependent capture coefficients, which may significantly affect the ISD shape and scaling properties. We also plan to consider more complex growth scenarios of reversible growth with non-zero detachment rates using a similar approach. Preliminary analysis of this work has shown that the ISD is time-invariant in terms of the variable ρ for any capture coefficients that increase faster than $s^{1/2}$ for large s, while the analytic scaling regimes in the FV form are rare. One simple case is the size-linear capture coefficients $\sigma \left(s\right)=s$, for which we have obtained the parameter-free ISD and the scaling function of the FV type. These analytic solutions are valid at any time rather than in the large time limit, and do not even require infinitely large Λ. The obtained VF scaling function does not vanish at x=0 and is monotonically decreasing. It has been demonstrated that the ISD shapes are very different in terms of invariant versus natural variables. For the size-linear capture coefficients, the ISD over invariant size is mono-modal and has a sharp maximum at the most representative size, while the ISD over natural size is monotonically decreasing and broad. We have also shown that introduction of the invariant size in the irreversible growth models enables one to safely neglect kinetic fluctuations in the continuum RE and operate with a time-invariant ISD regardless of the particular form of the capture coefficients. We hope that these results will be useful for modeling the epitaxial growth kinetics of different nanoclusters and for the general understanding of the scaling properties of ISDs.

## Figures and Tables

**Figure 1 nanomaterials-15-00396-f001:**
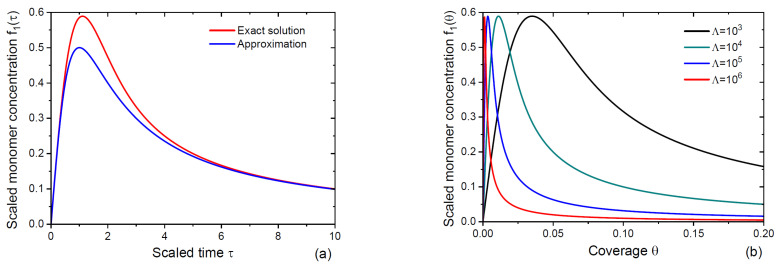
(**a**) Scaled monomer concentration $f_{1}=\sqrt{\Lambda }n_{1}$ versus scaled time τ as given by Equations (29) (exact solution) and (30) (approximation). (**b**) Scaled monomer concentration $f_{1}\left(\sqrt{\Lambda }\theta \right)$ given by Equation (29) versus coverage θ at different Λ shown in the legend. The monomer concentration, and hence the nucleation rate, become sharper and more asymmetric for larger Λ.

**Figure 2 nanomaterials-15-00396-f002:**
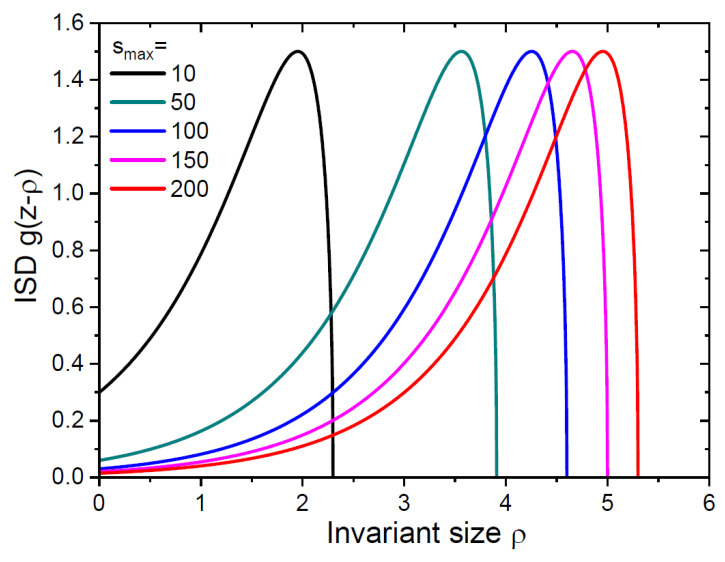
Continuum ISD over invariant size ρ at different $s_{max}$ shown in the legend. Curves are obtained from Equation (33) at α=3 and different $z=lns_{max}$.

**Figure 3 nanomaterials-15-00396-f003:**
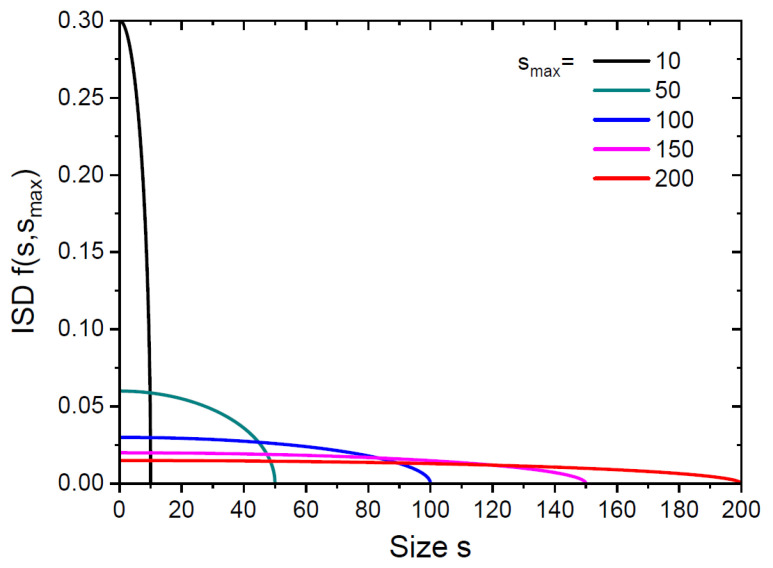
Continuum ISD over natural size s, obtained from Equation (34) at α=3 for the same $s_{max}$ as in Figure 2.

**Figure 4 nanomaterials-15-00396-f004:**
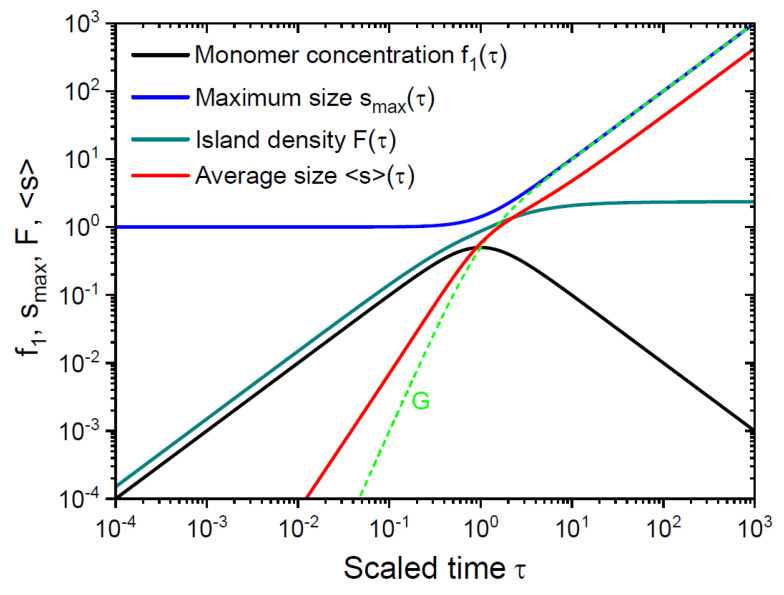
Monomer concentration, maximum size, island density and average size versus time. Dashed line shows the total number of monomers in the islands.

**Figure 5 nanomaterials-15-00396-f005:**
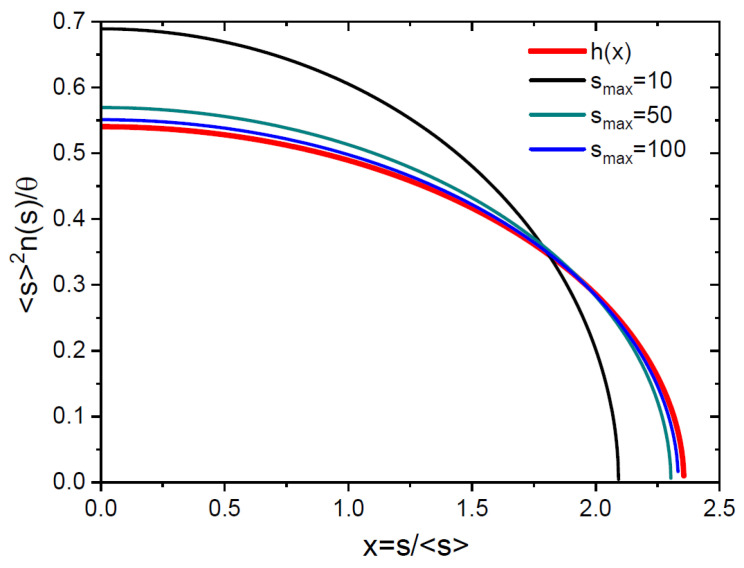
Scaled ISDs versus scaled size $s/\langle s\rangle$ at different $s_{max}$ shown in the legend. The ISDs rapidly converge to the universal scaling function given by Equation (39).

## Data Availability

Data are contained within the article.

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
