# Peer review of "A General Solution to the Continuum Rate Equation for Island-Size Distributions: Epitaxial Growth Kinetics and Scaling Analysis"

_nanomaterials, 2025, doi:10.3390/nano15050396_

Round 1
Reviewer 1 Report
Comments and Suggestions for Authors
This paper is the latest in a series published by the author on island-size distributions during submonolayer epitaxy guided by the Family-Vicsek scaling law. This program appears to be motivated by the original work in Refs. 16 and 22. The contribution of the author is to obtain solutions for various species subject to assumptions about the capture numbers, which are most difficult part of the rate equation formulation.
The best way to test this, and other theories of submonolayer epitaxy is to carry out extensive kinetic Monte Carlo simulations. Such simulations make no assumptions about capture numbers and any quantity, including effective capture numbers, as done in Ref.27, can be calculated. The authors is clearly aware of this, but I think that is the only way to verify his theory.
This manuscript can be published in its present form. The author is aware of all prior work, so his work is placed in the context of this work.
One possible typo: The caption to Fig. 5 refers to Eq. (38). This doesn’t seem correct.
Author Response
Point 1: One possible typo: The caption to Fig. 5 refers to Eq. (38). This doesn’t seem correct.
Response: Thank you for noting this error, which is now corrected:
The ISDs rapidly converge to the universal scaling function given by Equation (39).
Reviewer 2 Report
Comments and Suggestions for Authors
The paper is devoted to study continuous rate equations for island growth. Analytic solutions are found and the scaling properties analyzed. In my opinion, the paper is suitable for publication in Nanomaterials. I have only minor technical comments.
The use of commas in the equations is not unified. For example, commas in equations (1) and (3) are different. This is especially confusing in equation (9), where comma looks like an additional prime in (s´)´. It would be easier to follow the equations if commas would be like in equation (1), and some space added between the expression and comma.
Why equations in lines 78 and 80 are bold? There is no special reason for that.
Line 86: instead of “from” it should be “form”
Some text seems missed at or after line 108.
“ln” in equation (26) and “erf” in equation (29) should be roman (not italic).
Author Response
Point 1: The use of commas in the equations is not unified. For example, commas in equations (1) and (3) are different. This is especially confusing in equation (9), where comma looks like an additional prime in (s´)´. It would be easier to follow the equations if commas would be like in equation (1), and some space added between the expression and comma.
Response: This is corrected as requested.
Point 2: Why equations in lines 78 and 80 are bold? There is no special reason for that.
Response: Bold font is removed in these lines and elsewhere.
Point 3: Line 86: instead of “from” it should be “form”
Response: This typo is corrected:
Choosing ?1=1 is justified because the form given by Equations (1) is resumed …
Point 4: Some text seems missed at or after line 108.
Response: Thank you, this sentence is now completed:
In the limit of Λ→∞ typically considered for the FV scaling behavior, the islands rapidly reach large sizes in the pre-coalescence growth stage at modest coverages ?≪1.
Point 5: “ln” in equation (26) and “erf” in equation (29) should be roman (not italic).
Response: This is corrected as requested.